# Adolescent mental well-being, religion and family activities: a cross-sectional study (Northern Ireland Schools and Wellbeing Study)

Jordan Bamford ![ORCID],[1] Gerard Leavey ![ORCID],[2] Michael Rosato,[2] Natalie Divin,[3] Gavin Breslin,[3] Dagmar Corry[4]

¹Division of Psychology and Mental Health, The University of Manchester, Manchester, UK
²Bamford Centre for Mental Health and Wellbeing, University of Ulster, Coleraine, UK
³School of Psychology, Ulster University, Coleraine, UK
⁴School of Psychology, Queen's University Belfast, Belfast, UK

**Correspondence to**
Professor Gerard Leavey;
g.leavey@ulster.ac.uk

## ABSTRACT

**Objectives** In this study, we seek to explore the relationship between adolescent mental well-being, religion and family activities among a school-based adolescent sample from Northern Ireland.

**Setting** The Northern Ireland Schools and Wellbeing Study is a cross-sectional study (2014–2016) of pupils in Northern Ireland aged 13–18 years.

**Participants** 1618 adolescents from eight schools participated in this study.

**Outcomes measures** Our primary outcome measure was derived using the Warwick-Edinburgh Mental Wellbeing Scale. We used hierarchical linear regression to explore the independent effects of a range of personal/social factors, including religious affiliation, importance of religion and family activities.

**Results** In fully adjusted models, older adolescents and females reported lower mental well-being scores—for the year-on-year increase in age β=−0.45 (95% CI=−0.84, −0.06), and for females (compared with males) β=−5.25 (95% CI=−6.16, −4.33). More affluent adolescents reported better mental well-being. No significant differences in mental well-being scores across religious groups was found: compared with Catholics, Protestant adolescents recorded β=−0.83 (95% CI=−2.17, 0.51), other religious groups β=−2.44 (95% CI=−5.49, 0.62) and atheist adolescents β=−1.01 (95% CI=−2.60, 0.58). The importance of religion in the adolescents' lives was also tested: (compared with those for whom it was not important) those for whom it was very important had better mental well-being (β=1.63: 95% CI=0.32, 2.95). Higher levels of family activities were associated with higher mental well-being: each unit increase in family activity produced a 1.45% increase in the mental well-being score (β=0.78: 95% CI=0.67, 0.90).

**Conclusions** This study indicates that non-religious adolescents may have lower mental well-being scores when compared with their more religious peers, irrespective of religious denomination. This may relate to both a sense of lack of firm identity and perceived marginalisation. Additionally, adolescents with poor family cohesion are more vulnerable to poor mental well-being.

---

### STRENGTHS AND LIMITATIONS OF THIS STUDY

⇒ This study uses well-validated instruments to determine personal and social factors associated with mental well-being.
⇒ This study uses a representative sample of adolescents from a range of schools.
⇒ A limitation is that the study is cross-sectional, and therefore causality cannot be demonstrated.

---

## INTRODUCTION
### Adolescent mental health

Worldwide, mental disorders—particularly mood disorders—among children and adolescents worldwide may be increasing, something noted even before the recent COVID-19 pandemic.[1–5] In England, one National Health Service (NHS) survey found that 14.4% of 11–16 year olds reported a mental disorder,[5] and evidence suggests increased demand over the last decade for specialist mental health interventions for children and adolescents.[6] Additionally, there is evidence of increasing self-harm among young people, one study shows a 68% increase among girls aged 13–16 years since 2011.[7] In the UK, reported suicide has increased in those less than 18 years of age, mainly driven by an increase in females aged 16 years and males aged 17 years.[8]

Adolescence is a critical life-course period for health, with consequent mental health problems having a much greater long-term cost than those developed during adulthood.[9–11] The brain is particularly sensitive to stressors during this time.[12] Social isolation, self-esteem and connectedness are important determinants of adolescent mental health.[13] Associations between depressive symptoms and social media use have been identified, and this apparent vulnerability is greater for girls than boys.[14] Furthermore,

among adolescents, sex, social deprivation, child-in-need status, ethnicity and age all predict poor mental health outcomes.[15 16]

## Mental well-being, religion and family activities

Mental well-being is the positive aspect of mental health, relating to psychological functioning, life-satisfaction and ability to develop and maintain mutually beneficial relationships.[17] The dual-continua model of mental health suggests that mental illness and mental well-being are not extreme ends of a single spectrum but rather they reflect distinct continua which influence one another via complex interrelationships.[18] Among adults, better mental well-being is associated with reduced risks of developing mental illness and improved chance of recovery from mental illness.[19 20] Evidence suggests that mental well-being is positively associated with a broad range of positive mental and physical health outcomes among adolescents in the UK.[21]

Evidence suggests that greater religiosity (mainly religious service attendance) is associated with higher well-being among adults,[22 23] although this evidence is derived mainly from predominantly Christian groups. One natural experiment suggests that greater involvement in religious activity (Ramadan fasting) increases well-being among Muslims.[24] Evidence suggests religiosity and spirituality are beneficial for adolescents: correlating with positive health attitudes and behaviours[25] and having a positive effect on psychological outcomes including risk behaviour, depression, well-being and self-esteem.[26] It is thought that attendance at religious services can bolster mental well-being by reducing loneliness, providing social support and engagement with community[27 28]; and evidence from the UK also indicates that more frequent attendance improves well-being among adults.[29] However, religion can be damaging to mental well-being,[30] particularly for vulnerable groups such as lesbian, gay or bisexual adolescents.[31] Another key determinant of mental well-being is within-family group activities, which evidence suggests is both an independent predictor of adolescent well-being,[32] and a measure of family cohesion. The family has both a role in moderating risk factors for poor child development[33] and influences the building of resilience.[34]

## The context in Northern Ireland

Adolescent mental health in Northern Ireland (NI) is a public health concern following increased suicidality among young people.[21 35–37] Previous research suggests that post-conflict generations who have experienced violence in their communities may be at higher risk of depression, psychotic symptoms or substance misuse.[38] Religious affiliation is of continuing social importance since the putative ending of the civil conflict (The Troubles). The most recent census data (from 2021) shows that NI comprises a small majority Catholic (45.7%) and large minority Protestant group (43.5%), with 1.5% identifying as other non-Christian, and the remaining 9.3% of the population identifying as not belonging to any religion—with this latter group increasing from 5.6% in 2011.[39] There is evidence that minority religious and atheist groups report poorer mental health and increased suicidality.[36] Loneliness and poverty may also be associated with poor mental health in this population.[40]

In this study, we explore the relationship between adolescent mental well-being, religion and family cohesion. Such findings offer useful insights to inform child and adolescent mental health services and guide policy.

## METHODS

### Study design and setting

This is a cross-sectional study (2014–2016) of pupils in NI aged 13–18 years, based on a questionnaire and linked school administrative records data.[36 41] Eight schools were selected from a database of post-primary schools based on locale (urban/rural) and school type (grammar or secondary modern). While grammar school enrolment is based on selection (through examination), it is generally acknowledged to have a strong middle-class bias. In NI, 43% of post-primary children attend grammar schools.[42 43] As part of the study protocol, parents received a letter explaining the study, with an opt-out form for those requesting non-participation. All pupils provided written informed consent. Only five parents or children declined to participate. School absence was between 5% and 8% across the selected schools. Pupils completed the questionnaires in class groups following procedures described previously.[44] A copy of the questionnaire is included as online supplemental file 1. Our sample comprised of 1618 young people who provided complete responses to the questionnaire.

### Measures

#### Outcome variable

Mental well-being: the Warwick-Edinburgh Mental Well-being Scale is a widely used measures of mental well-being,[45 46] in both NHS and private health services. Overall, it is deemed a valid and reliable tool for measuring well-being in diverse populations.[47 48] It comprises 14 items covering psychological functioning and subjective well-being, each a positively worded question on a five-point Likert scale. The responses are summed to provide a total score which ranges between 14 and 70, with higher scores indicating better mental well-being.

#### Measures of religion

Religious group: participants indicated which religious group they identified with—summarised as Catholic, Protestant, other faith group or atheist/agnostic.

Importance of religion: participants were asked about 'how important is religion to your family'—with responses very important, moderately important or not important.

#### Family activities

To assess family cohesion, we used the Family Activities scale[49]—six questions, each a five-point Likert scale (with

responses 1=never to 5=everyday). The responses are summed, with high scores indicating an integrated family life.

## Confounders

We controlled for age (continuous) and ethnicity (white European, other ethnic group). Sex (male, female)—rather than gender—was recorded, as requested by school principals. We controlled for school type (grammar, non-grammar) and family affluence—the latter measured using the family affluence scale which quantifies socio-economic inequity. This measure is reliable, sensitive and validated for differentiating levels of affluence.[50] It was used in analysis as a continuous variable (range=0 to 9), with higher scores indicating greater affluence. We also controlled for the locale of the school (rural, urban).

## Analysis

Characteristics of the population were determined, and the relationship with characteristics of the population and mental well-being examined, with mean and SD presented. To assess for significant differences between means, we used t-tests and analysis of variances (for binary or categorical variables, respectively). For continuous variables, means and SD are presented. We used hierarchical linear regression to explore the influence of religion, religious importance and family activities on well-being—and present results as beta coefficient, associated 95% CIs and p values.

## Patient and public involvement

None.

## RESULTS

Population characteristics are presented in table 1. The sample included adolescents aged 13 to 18 years, with over 90% aged 16 years or younger. This sample was mainly female (62.7%); predominantly white European (94.7%), with 57.5% attending non-grammar schools. With family affluence, the mean score was 6.0 (SD 1.7); while for family activities the mean was 16.6 (SD 3.8). Most adolescents identified as Catholic (60.4%) and 10.3% identified as being atheist/agnostic; and over 70% said that religion was either moderately or very important to their family. In this sample, 60.3% of adolescents attended school in an urban area.

The mean score for mental well-being over the whole sample was 47.7 (SD 9.7). We identified no significant difference across age group and ethnic group with regard to mental well-being (unadjusted for other confounding predictors). We identified that across the sexes, religious groups, religious importance to family, rurality and school type, there were significant differences in mean scores for mental well-being (unadjusted for other confounding variables). We identified that females had on average lower mental well-being scores compared with males, 45.7 (SD 9.9) and 51.0 (SD 8.4), respectively. Lower well-being

scores were found among atheist/agnostic adolescents and those adolescents whose families did not consider religion to be important. Adolescents in grammar schools had lower on average mental well-being scores, as too did those adolescents attending rural schools. We identified that higher scores for both the Family Act Scale and the Family Influence Scale were associated with greater well-being scores for adolescents.

Table 2 presents results for four incrementally adjusted models (M1 to M4) examining the association between well-being and the selected risk factors. In M1, both older and female adolescents record significantly poorer mental well-being. More affluent adolescents recorded better mental well-being. Neither grammar school status, rurality or ethnicity were significantly associated with predicting mental well-being.

The addition of religious group (M2) did not materially influence the already established associations of sex, age and affluence with mental well-being. However, atheist adolescents (compared with Catholics) reported significantly poorer mental well-being (β=−2.98: 95% CI=−4.57, −1.40). No significant differences between the reference Catholics and Protestant or adolescents from other religious groups ere identified.

With the inclusion of importance of religion (M3), the established associations between well-being and age, sex and family affluence persisted, and while the poorer well-being associated with non-religious adolescents was again evident, its impact on poorer well-being was reduced—from β=−2.98 (95% CI=−4.57, −1.40) to β=−2.06 (95% CI=−3.73, −0.40) in M2 and M3, respectively. Adolescents whose family considered religion to be either moderately or very important were (compared with those for whom religion was of no importance), reported significantly better well-being, for example, β=2.56 (95% CI=1.19, 3.94), seen in those for whom religion was very important.

Finally, M4—adding family activity levels—shows the fully adjusted model. Again, the influence of sex and age persists, as too does family affluence (though with reduced beta coefficients). Inclusion of family activity levels also: (a) obviates any association between all religious denomination and well-being—including the non-religious group; and (b) constrains any beneficial effects of the importance of religion, and only those adolescents where religion was very important retain any beneficial effect (β=1.63: 95% CI=0.32, 2.95). Family activity levels are associated with significant higher mental well-being—β=0.78 (95% CI=0.67, 0.90), a per-unit increase of 1.45% well-being.

## DISCUSSION

In this study, examining predictors of mental well-being in a school-based sample of adolescents, we found that in families where religion was not important, there were poorer adolescent well-being outcomes scores compared with their peers for whom religion was very important. This relationship has been identified in other studies which explored

**Table 1** Sample characteristics and unadjusted distribution of well-being

| | Whole sample | WEMWBS score | |
| | % (n) | Mean (SD) | Significance |
|---|---|---|---|
| **Age** | | | |
| 13 | 11.5 (186) | 50.1 (8.5) | 0.203 |
| 14 | 30.4 (492) | 48.4 (9.5) | |
| 15 | 25.7 (416) | 47.1 (10.3) | |
| 16 | 24.4 (395) | 46.9 (9.8) | |
| 17 | 7.2 (116) | 46.0 (9.4) | |
| 18 | 0.8 (13) | 47.6 (8.6) | |
| **Sex** | | | |
| Male | 37.3 (603) | 51.0 (8.4) | <0.001 |
| Female | 62.7 (1015) | 45.7 (9.9) | |
| **Ethnicity** | | | |
| White European | 94.7 (1532) | 47.7 (9.8) | 0.188 |
| Other | 5.3 (86) | 47.9 (8.7) | |
| **Religious grouping** | | | |
| Catholic | 60.4 (978) | 47.8 (9.6) | <0.001 |
| Protestant | 27.1 (439) | 48.4 (9.4) | |
| Other | 2.1 (34) | 46.1 (11.3) | |
| Atheist/agnostic | 10.3 (167) | 45.4 (10.8) | |
| **Importance of religion** | | | |
| Very | 19.8 (321) | 49.3 (10.1) | 0.03 |
| Moderately | 51.2 (829) | 48.0 (9.1) | |
| Unimportant | 28.9 (468) | 46.1 (10.3) | |
| **School type** | | | |
| Grammar | 42.5 (688) | 46.7 (10.1) | 0.004 |
| Non-grammar | 57.5 (930) | 48.4 (9.4) | |
| **Rurality** | | | |
| Rural | 39.7 (643) | 47.0 (10.0) | <0.001 |
| Urban | 60.3 (975) | 48.8 (9.3) | |
| **Continuous variables** | | | |
| | **Mean (SD)** | **Beta coefficient** | **P value** |
| **Family activities** | | | |
| Range 6–30 | 16.6 (3.8) | 0.88 | <0.001 |
| **Family affluence** | | | |
| Range 0–9 | 6.0 (1.7) | 0.63 | <0.001 |
| **Mental well-being** | | | |
| Range 14–70 | 47.7 (9.7) | – | – |

WEMWBS, Warwick-Edinburgh Mental Wellbeing Scale.

the influence of religious importance and its impact independent of religious affiliation.[51] We initially identified that atheist/agnostic adolescents had lower mental well-being scores compared with other adolescent groups (catholic/protestant/other groups); however, when we accounted for differences in family activities, difference of mental well-being scores across religious groups were insignificant. Thus, where

previous research has found non-religious groups having better mental health outcomes, this is not evident among our sample.[52] Previous research using this sample found evidence of increased suicidality and mental illness associated with non-religious adolescents[36]—this apparent vulnerability may relate to social and psychological challenges associated with both being different and a felt lack of firm identity. This

**Table 2** Relationship between mental well-being and included demographic and family characteristics

| | M1 Beta coefficient (95% CI) | M2 Beta coefficient (95% CI) | M3 Beta coefficient (95% CI) | M4 Beta coefficient (95% CI) |
|---|---|---|---|---|
| Age† | **−0.82 (−1.23, −0.42)**\*\* | **−0.75 (−1.16, −0.34)**\*\* | **−0.76 (−1.17, −0.35)**\*\* | **−0.45 (−0.84, −0.06)**\* |
| Sex | | | | |
| Female (ref=male) | **−5.21 (−6.17, −4.25)**\*\* | **−5.41 (−6.37, −4.44)**\*\* | **−5.38 (−6.34, −4.41)**\*\* | **−5.25 (−6.16, −4.33)**\*\* |
| Ethnicity | | | | |
| (ref=white European) | | | | |
| Other | −0.64 (−2.66, 1.39) | −0.23 (−2.27, 1.81) | −0.32 (−2.35, 1.72) | −0.29 (−2.22, 1.65) |
| School type | | | | |
| (ref=grammar) | | | | |
| Non-grammar | 0.00 (−1.37, 1.38) | 0.79 (−1.00, 2.58) | 0.94 (−0.85, 2.73) | 1.23 (−0.46, 2.93) |
| Family affluence† | **0.76 (0.48, 1.03)**\*\* | **0.75 (0.48, 1.02)**\*\* | **0.70 (0.43, 0.97)**\*\* | **0.41 (0.14, 0.67)**\* |
| Locale of residence | | | | |
| Rural (ref=urban) | 0.99 (−0.31, 2.29) | 0.31 (−1.19, 1.80) | 0.13 (−1.37, 1.63) | −0.61 (−2.04, 0.82) |
| Religion | | | | |
| (ref=Catholic) | | | | |
| Protestant | | −0.82 (−2.23, 0.59) | −0.60 (−2.01, 0.81) | −0.83 (−2.17, 0.51) |
| Other | | −2.95 (−6.18, 0.28) | −3.05 (−6.28, 0.16) | −2.44 (−5.49, 0.62) |
| Atheist | | **−2.98 (−4.57, −1.40)**\*\* | **−2.06 (−3.73, 0.40)**\* | −1.01 (−2.60, 0.58) |
| Importance of religion | | | | |
| (ref=not important) | | | | |
| Moderately | | | **1.49 (0.37, 2.61)**\* | 0.91 (−0.16, 1.9) |
| Very | | | **2.56 (1.19, 3.94)**\*\* | **1.63 (0.32, 2.95)**\* |
| Family activities† | | | | **0.78 (0.67, 0.90)**\*\* |

Data represents beta coefficients (and 95% CIs) from four incrementally developed regression models.
\*<0.05; \*\*<0.001.
†Variable treated as continuous.
figures in bold represents statistical significance

experience of marginalisation associated with non-religious groupings has been found to contribute to poorer mental health outcomes in the USA.[53 54]

In this sample, females (compared with males) experienced poorer well-being: female vulnerability has been identified elsewhere: one study of Mexican adolescents found that post-menarche females recorded increased depressive symptoms, externalising problems and negative body image[55]; and another study identified that older female adolescents reported lower levels of self-compassion and lower emotional well-being.[56] Male/female differences in well-being have been related to societal cultural attitudes and inequity surrounding gender.[57]

In this study, more cohesive family activity levels were key to better mental well-being. Adolescents with secure attachments and cohesive families have been found to engage less in high-risk behaviours, have fewer mental health problems, enhanced social skills and better coping strategies.[58] Some evidence suggests that parenting style and, by implication, family activity levels play a key role in emerging adult mental health, particularly for females.[59]

In the fully adjusted model, high levels of affluence were significantly associated with better well-being. In an earlier study using this, data affluence had little influence on either mental health outcomes or trust in GPs.[36] Family affluence may therefore be associated with mental well-being, but not necessarily mental health outcomes. Other research has shown that mental well-being and mental illness have distinct and different correlates and that actual and perceived socioeconomic circumstance may correlate differently from each other.[60] Other research has shown that socioeconomic position was associated with self-harm among adolescents, mainly for girls,[61] and a systematic review found lower socioeconomic status associated with mental health problems.[62]

### Strengths and limitations

This study uses well-validated instruments to determine personal and social factors associated with mental well-being. This sample of adolescents from eight schools is relatively representative, with similar proportions of adolescents in grammar school education,[43] and a similar proportion of atheist/agnostic adolescents compared with the general NI population.[39] While we note that Catholics are over-represented in this sample (60% from a Catholic background, compared with 45.7% in the general population)[39]—current demographic trends in post-primary school enrolment record higher proportions of adolescent Catholics: in 2021/2022, 50.2% of the grammar school population (32 990/65 313) and 51.7% (44 795/86 588) of the non-grammar school population were Catholic.[43] This age-related redistribution of Catholics in relation to Protestants in NI mirrors current local demographic trends.[63] Furthermore, our sample contains 5.3% of adolescents who do not belong to white European ethnicity. Census data

shows that 3.4% of the population are from minority ethnic groups in NI and tend to be younger which is in keeping with our sample.[64]

A limitation is that the study is cross-sectional, and therefore causality cannot be inferred. Further we recognise that the use of self-reported data has limitations, with social desirability bias possibly impacting on the results. Finally, as previously mentioned, we measured sex (not gender) at the request of school principals—we therefore acknowledge that this study cannot explore the influence of gender identity problems on mental well-being.

### CONCLUSION

This study suggests that religious practice and family activities are important determinants of adolescent well-being. We highlight vulnerabilities associated with adolescents for whom religion is not important, in relation to the experience of marginalisation and perceived lack of a firm identity. Additionally, the study suggests that less cohesive family units may also present vulnerabilities which can impact poor adolescent mental well-being. Future policy should be mindful of these personal and social factors, and child and adolescent mental health services in NI should consider the increased vulnerability of non-religious youth.

**Contributors** GL: design and principal investigator; GL is guarantor for the study conduct, findings and data access; DC and ND: data collection; JB, GL and MR: data analysis; JB, GL, MR, DC, ND, GB: writing. All authors have agreed this manuscript and confirm that there are no past, current or potential conflicts of interest.

**Funding** This study was supported through Ulster University and the R&D Division of Northern Ireland Public Health Agency. The funders played no other role in the study and have not contributed to the analysis or writing up of the results which are the sole responsibility of the authors. Funding reference number: NA.

**Competing interests** None declared.

**Patient and public involvement** Patients and/or the public were not involved in the design, or conduct, or reporting, or dissemination plans of this research.

**Patient consent for publication** Not applicable.

**Ethics approval** This study involves human participants and was accorded a favourable opinion by the Ulster University Research Ethics Committee. Ethics approval number UREC20/16 NISAW. Participants gave informed consent to participate in the study before taking part.

**Provenance and peer review** Not commissioned; externally peer reviewed.

**Data availability statement** An anonymised, de-identified database can be made available on request. Not applicable.

**ORCID iDs**
Jordan Bamford http://orcid.org/0000-0003-4897-2039
Gerard Leavey http://orcid.org/0000-0001-8411-8919

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
