## [Reviewer comments · BMJ Open]

ARTICLE DETAILS

TITLE (PROVISIONAL)	Adolescent mental wellbeing, religion, and family activities: a cross-sectional study (Northern Ireland Schools and Wellbeing Study)
AUTHORS	Bamford, Jordan; Leavey, Gerard; Rosato, Michael; Divin, Natalie; Breslin, Gavin; Corry, Dagmar

VERSION 1 – REVIEW

REVIEWER	Kaufman, Caroline C McLean Hospital, Psychiatry
REVIEW RETURNED	08-Mar-2023

GENERAL COMMENTS	Thank you for the opportunity to review this paper describing a cross-sectional study of mental health, well-being, and religion among a sample of adolescents from Northern Ireland. This study responds well to worldwide concerns regarding child and adolescent mental health. However, authors should consider the following comments, questions, and suggestions prior to publication. Minor: -There appears to be a type in the heading "Mental Wellbeing and Religion" (p. 4)-Mental well-being is sometimes referred to as "Mental Wellbeing" or "Mental well-being." Authors should be consistent throughout the manuscript. Major: -Authors must clarify that they conducted a hierarchical linear regression. As written, it is not clear that authors did not simple conduct a multiple regression. It is only clear when the table is referenced.-Authors cite several studies demonstrating the relationship between religion and mental well-being (p. 4). However, many of these studies appear to be with adult samples, authors should specify which findings are from adult versus adolescent samples. In addition, authors should consider primarily highlighting research demonstrating this intersection among adolescents rather than adults given the study sample.-Do authors consider Catholicism to be a religious minority in Northern Ireland? Readers not familiar with The Troubles and related history in the area may not be aware of which religious groups are considered minorities in Northern Ireland.-The proportion of Catholics in the sample (approximately 60%) is higher than the percentage in the entire country. What do authors make of the somewhat overrepresentation of Catholics in their sample?
---

	-Authors assert that they measure "gender" in their study, but report frequencies of males and females. As "male" and "female" are used to describe sex, authors should change their description to indicate that they measured "sex." In addition, it is notable that the sample did not include any intersex participants. Were participants allowed to choose other options other than female or male? -Did authors explore interactions across gender and other significant predictors (e.g., religious importance, family affluence). Examining interactions may provide additional information regarding how these factors predict well-being. -It is not clear why authors claim that their sample is representative given that the authors do not provide statistics regarding the demographics of the entire country. Authors should add this information and/or explain why they believe their sample is representative. -Authors should consider expanding the limitations and strengths section. Specifically, authors should identify potential limitations in self-report data (e.g., social desirability bias) as well as other potential limitations (e.g., binary measurement of sex, lack of measurement of gender identity). -Table 1 includes p values, but it is not clear what these p values refer to. Is it the significance level in the final stage of the hierarchical regression? If so, this can be removed from the table since it should be reported in either Table 2 or the text of the manuscript.
--	---

REVIEWER	Pedroso, Janari Universidade Federal do Pará, Programa de Pós-Graduação em Teoria e Pesquisa do Comportamento
REVIEW RETURNED	13-Mar-2023

GENERAL COMMENTS	In the introduction, the concepts of mental health and mental well-being are broad, for example, it would be important to focus on psychological functioning and subjective well-being. Failed to explain about family bonding. In the discussion, the authors address some points that weaken the analysis, for example on page 6, line 46 to 54: Female vulnerability was identified elsewhere, a study that examined adolescents in Mexico found that women after menarche had increased depressive symptoms, externalization problems and negative body image, still on the same page in line 56 to 60 state that family bonding was a determining factor of mental well-being, with greater bonding predicting greater mental well-being, but it is discussed with attachment ver. The statements are confusing as female vulnerability and attachment are not in the results. It would be important to discuss with the results analyzed in table 2.
---

VERSION 1 – AUTHOR RESPONSE

Reviewer: 1

Dr. Caroline C Kaufman, McLean Hospital, Harvard Medical School

Comments to the Author:

Thank you for the opportunity to review this paper describing a cross-sectional study of mental health, well-being, and religion among a sample of adolescents from Northern Ireland. This study responds

well to worldwide concerns regarding child and adolescent mental health. However, authors should consider the following comments, questions, and suggestions prior to publication.

Minor:

-There appears to be a type in the heading "Mental Wellbeing and Religion" (p. 4)
Corrected.

-Mental well-being is sometimes referred to as "Mental Wellbeing" or "Mental well-being." Authors should be consistent throughout the manuscript.

Thank you, we have updated with consistent use of wellbeing.

Major:

-Authors must clarify that they conducted a hierarchical linear regression. As written, it is not clear that authors did not simple conduct a multiple regression. It is only clear when the table is referenced.

We have updated both the abstract and the methodology section to reflect the clear use of hierarchical linear regression, further we have updated the heading of table 2.

-Authors cite several studies demonstrating the relationship between religion and mental well-being (p. 4). However, many of these studies appear to be with adult samples, authors should specify which findings are from adult versus adolescent samples. In addition, authors should consider primarily highlighting research demonstrating this intersection among adolescents rather than adults given the study sample.

Thank you for this feedback. We have updated the background section on mental wellbeing and religion, highlighting whether referenced evidence is from adult or adolescent populations, further we have added further reference material focussing on adolescent populations such as:

Rew, L., & Wong, Y. J. (2006). A systematic review of associations among religiosity/spirituality and adolescent health attitudes and behaviors. *Journal of adolescent health, 38*(4), 433-442.

Yonker, J. E., Schnabelrauch, C. A., & DeHaan, L. G. (2012). The relationship between spirituality and religiosity on psychological outcomes in adolescents and emerging adults: A meta-analytic review. *Journal of adolescence, 35*(2), 299-314.

-Do authors consider Catholicism to be a religious minority in Northern Ireland? Readers not familiar with The Troubles and related history in the area may not be aware of which religious groups are considered minorities in Northern Ireland.

Thank you for raising this important point. We have updated the background section with details from the most recent census in Northern Ireland, indicating that Catholicism in Northern Ireland is a small majority religious group and highlighted that religious affiliation is of ongoing social importance since the end of conflict.

-The proportion of Catholics in the sample (approximately 60%) is higher than the percentage in the entire country. What do authors make of the somewhat overrepresentation of Catholics in their sample?

Thank you. We have updated the paper to ensure that it is clear that our sample has a degree of over representation of catholic adolescents (described in the strengths and limitations section) and relate this to demographic shifts.

-Authors assert that they measure "gender" in their study, but report frequencies of males and females. As "male" and "female" are used to describe sex, authors should change their description to indicate that they measured "sex." In addition, it is notable that the sample did not include any intersex participants. Were participants allowed to choose other options other than female or male?

This comment is appreciated, and we have updated the terminology used to ensure reflection of what is being described with consistent use of 'sex' as opposed to gender, further we have updated the methodology to indicate that sex not gender was measured at the request of school principals.

-Did authors explore interactions across gender and other significant predictors (e.g., religious importance, family affluence). Examining interactions may provide additional information regarding how these factors predict well-being.

Interaction with gender and significant predictors on mental wellbeing were not explored in this analysis. There is a dearth of research looking at religiosity among adolescents in Northern Ireland and we wanted the focus of the analysis to be on the role of importance of religion, church attendance and family activities. We feel that while exploring for gender specific interactions is worthwhile and we may do in the future, it would dilute our present study and was not linked to our initial aim.

-It is not clear why authors claim that their sample is representative given that the authors do not provide statistics regarding the demographics of the entire country. Authors should add this information and/or explain why they believe their sample is representative.

As above, we have updated.

-Authors should consider expanding the limitations and strengths section. Specifically, authors should identify potential limitations in self-report data (e.g., social desirability bias) as well as other potential limitations (e.g., binary measurement of sex, lack of measurement of gender identity).

Thank you. We have updated the limitations and strengths section with consideration of these important points.

-Table 1 includes p values, but it is not clear what these p values refer to. Is it the significance level in the final stage of the hierarchical regression? If so, this can be removed from the table since it should be reported in either Table 2 or the text of the manuscript.

Thank you. We have updated both the table title and in the text of the results section to clarify that we explored how the sample characteristics in this study relate to mental wellbeing, and that tests of significant differences were unadjusted for other confounding predictors.

Reviewer: 2

Dr. Janari Pedroso, Universidade Federal do Pará

Comments to the Author:

In the introduction, the concepts of mental health and mental well-being are broad, for example, it would be important to focus on psychological functioning and subjective well-being.

We have sought to provide a definition of mental wellbeing, and provided more evidence for the impact of mental wellbeing on mental illness and recovery with the below references:

Keyes, C. L., Dhingra, S. S., & Simoes, E. J. (2010). Change in level of positive mental health as a predictor of future risk of mental illness. *American journal of public health, 100*(12), 2366-2371.

Iasiello, M., van Agteren, J., Keyes, C. L., & Cochrane, E. M. (2019). Positive mental health as a predictor of recovery from mental illness. *Journal of Affective Disorders, 251*, 227-230.

We identify mental wellbeing in the dual continua model of health

Further, we have included more about the impact of religiosity on psychological outcomes such as risk behaviour, depression and self-esteem

Failed to explain about family bonding.

Thank you for this comment. In order to improve clarity, we have changed the title of the article and use of the phrase 'family bonding' to 'family activities' as this is more representative of what was measured. Further, we now added more information in the background information section on the justification for the inclusion of this variable, and current understanding of why it is an important predictor of wellbeing.

In the discussion, the authors address some points that weaken the analysis, for example on page 6, line 46 to 54: Female vulnerability was identified elsewhere, a study that examined adolescents in Mexico found that women after menarche had increased depressive symptoms, externalization problems and negative body image, still on the same page in line 56 to 60 state that family bonding was a determining factor of mental well-being, with greater bonding predicting greater mental well-

being, but it is discussed with attachment ver. The statements are confusing as female vulnerability and attachment are not in the results. It would be important to discuss with the results analyzed in table 2.

We have drawn from other examples in research with adolescent samples to explain our finding that female adolescents in our study had lower mental wellbeing compared to males. We identify that two other studies found that adolescent females were more at risk of lower mental wellbeing and relate this to gender inequality and cultural attitudes towards gender. We relate our findings that family activities are important for mental wellbeing to attachment theory and other findings that cohesive families can be protective for mental wellbeing and mental illness. While attachment and cultural attitudes were not directly measured in our study, we use these concepts to explain our findings. We feel we have comprehensively examined the findings from table 2.